# Domain-Specific Data Synthesis for LLMs via Minimal Sufficient Representation Learning

## Abstract

Large Language Models have demonstrated remarkable progress in general-purpose capabilities and can achieve strong performance in specific domains through fine-tuning on domain-specific data. However, acquiring high-quality data for target domains remains a significant challenge. Existing data synthesis approaches follow a deductive paradigm, heavily relying on explicit domain descriptions expressed in natural language and careful prompt engineering, limiting their applicability in real-world scenarios where domains are difficult to describe or formally articulate. In this work, we tackle the underexplored problem of domain-specific data synthesis through an inductive paradigm, where the target domain is defined only through a set of reference examples, particularly when domain characteristics are difficult to articulate in natural language. We propose a novel framework, DOMINO, that learns a minimal sufficient domain representation from reference samples and leverages it to guide the generation of domain-aligned synthetic data. DOMINO integrates prompt tuning with a contrastive disentanglement objective to separate domain-level patterns from sample-specific noise, mitigating overfitting while preserving core domain characteristics. Theoretically, we prove that DOMINO expands the support of the synthetic data distribution, ensuring greater diversity. Empirically, on challenging coding benchmarks where domain definitions are implicit, fine-tuning on data synthesized by DOMINO improves Pass@1 accuracy by up to 4.63% over strong, instruction-tuned backbones, demonstrating its effectiveness and robustness. This work establishes a new paradigm for domain-specific data synthesis, enabling practical and scalable domain adaptation without manual prompt design or natural language domain specifications. *Code will be available upon publication.*

## 1 Introduction

Recent advances in Large Language Models (LLMs) (OpenAI, 2022; Gemini Team, 2024; Qwen et al., 2025; DeepSeek-AI, 2024) have revolutionized natural language processing, allowing for unprecedented performance in coding (Jain et al., 2024; Zhuo et al., 2025), mathematics (Cobbe et al., 2021; Ye et al., 2025), and various downstream tasks (Plaat et al., 2024; Zhang et al., 2024a; Nie et al., 2024). In real-world applications, the prevailing approach to downstream adaptation involves leveraging high-quality, domain-specific data for supervised instruction tuning (Ding et al., 2023; Zhang et al., 2024b). This approach rests on a crucial assumption that practitioners are able to obtain a sufficient number of annotated examples to explicitly characterize the target domain.

However, current data synthesis methods share a fundamental, yet often overlooked, limitation: they presuppose the existence of an explicit, human-articulable domain definition. These methods, whether based on Instruction Evolution (Wang et al., 2023a; Xu et al., 2024), Key-point-driven synthesis (Huang et al., 2024a;b), or System Prompts (Xu et al., 2025), all require translating domain knowledge into textual instructions. (Details of these related methods are shown in Appendix A.) This approach is analogous to **deductive reasoning**—starting from a general rule (the prompt) to generate specific instances. This paradigm breaks down when we consider how humans often learn. We excel at **inductive reasoning**: when confronted with a new concept—be it a genre of music, a style of art, or a type of logical puzzle—we do not begin with a formal, axiomatic definition. Instead, we learn by observing a handful of canonical examples and inducing the underlying rules, patterns, and boundaries of the domain. This raises a critical question for machine learning:

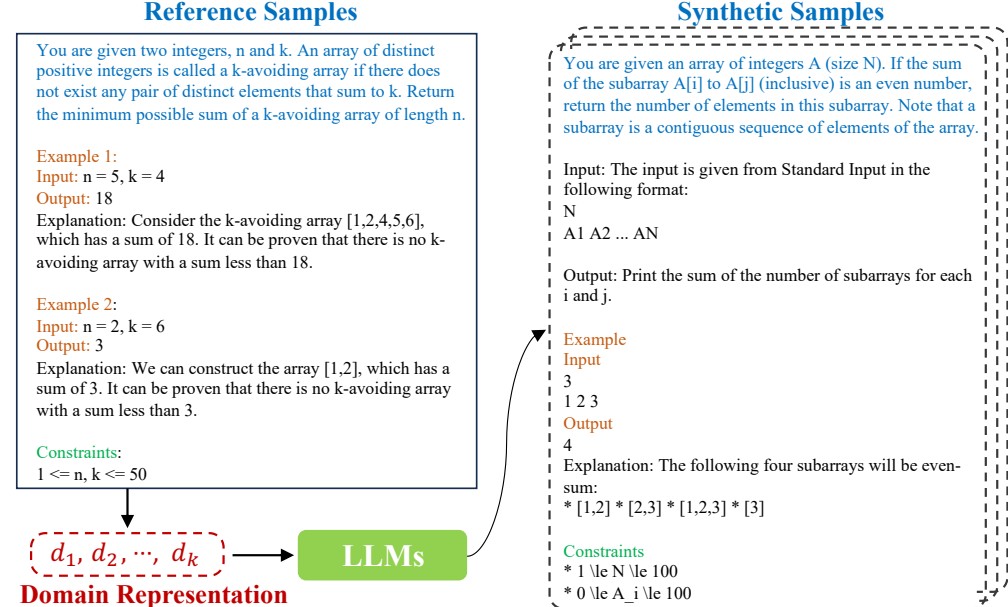

Figure 1: DOMINO encodes domain characteristics into latent embeddings derived from reference samples, which are then used to guide LLMs in generating new samples. While the domain characteristics are not explicitly defined in natural language, a comparison between reference and synthetic samples at least reveals a shared structural pattern (e.g., description, examples, constraints), even though the specific content differs, highlighting the DOMINO's ability to capture the domain's format.

> *"How can we synthesize domain-specific data when the target domain is defined implicitly by a set of reference examples, rather than explicitly in natural language?"*

This problem is not a niche concern but is central to practical domain adaptation. It arises in high-value scenarios such as emerging scientific fields where terminology is still fluid, rapidly evolving cultural trends like online slang, or with proprietary business logic that is confidential or poorly documented. In all these cases, the domain is governed by subtle conventions and emergent patterns that resist simple verbalization. While providing examples is easy, writing a formal definition is nearly impossible. A tempting but flawed approach is to simply use the available examples for supervised fine-tuning (SFT). With a limited dataset, however, SFT encourages the model to memorize superficial features of the training data rather than generalize the underlying domain principles (Chu et al., 2025). This leads to poor performance on novel, out-of-distribution tasks. This risk is not merely theoretical; our own experiments (Table 1) confirm that naive SFT on a reference set can fail to improve, or even degrade, results, highlighting its inadequacy for true domain adaptation.

The challenge, therefore, is not merely one of data quantity—a problem that synthesis can, in principle, solve by generating as many samples as desired—but of learning quality. To truly learn a domain from examples, a model must distinguish its core, generalizable principles from the idiosyncratic details of each sample. Our key insight is that this can be achieved by learning a minimal sufficient representation—a latent embedding that is sufficient to capture the essential characteristics of the domain while being minimal enough to discard irrelevant, sample-specific noise that leads to overfitting. As shown in Figure 1, we propose a framework that derives this minimal representation from reference samples and then uses it to guide an LLM in generating novel, diverse samples that adhere to the domain's induced principles.

We call our framework DOMINO (DOmain-specific data synthesis through MINimal sufficient representatiOn learning). Drawing on information-theoretic principles, DOMINO operates through two synergistic components. First, it uses prompt tuning to learn a continuous domain embedding that maximizes the likelihood of the reference examples, ensuring sufficiency. Second, it introduces a contrastive disentanglement mechanism that explicitly separates shared domain-level patterns from unique sample-level features, enforcing minimality. By optimizing a combined objective that enforces both data reconstruction fidelity and information minimality, DOMINO learns to abstract the domain's "first principles" from the examples provided. In summary, the primary contributions are:

- We formalize and tackle the problem of domain synthesis from implicit examples, mirroring human inductive reasoning. We develop DOMINO, a framework that learns a minimal sufficient representation by combining prompt tuning with contrastive disentanglement to capture essential domain patterns and promote generalization.

- Theoretically, we prove that DOMINO expands the support of the synthetic data distribution compared to baselines, ensuring greater diversity in generated examples.

- Empirically, we show DOMINO's effectiveness and robustness across diverse tasks. On coding tasks with implicit domains, it improves Pass@1 by up to 4.63% over strong instruction tuned models, and consistently outperforms baselines on both coding and instruction-following tasks.

## 2 METHODOLOGY

### 2.1 PROBLEM DEFINITION

We consider the problem of synthesizing data for a specific domain, denoted by $\mathcal{D}$, with a corresponding observation space $\mathcal{X}$. We assume access to a limited set of reference data within $\mathcal{X}$, that is, $\boldsymbol{X}^{(1:n)} = \{\boldsymbol{X}^{(1)}, \boldsymbol{X}^{(2)}, \cdots, \boldsymbol{X}^{(n)}\}$. Each $\boldsymbol{X}^{(i)}$ consists of both input and output sequences expressed as plain text (i.e., the data for supervised fine-tuning of LLMs). The size of the reference set, $n$, typically ranges from tens to hundreds of examples, a scale limited by high collection and annotation costs. Even within this range, distinct challenges require a carefully designed representation. When $n$ is on the lower end (e.g., tens of examples), the primary risk is overfitting to the idiosyncratic details of the few available samples. As $n$ grows larger (e.g., hundreds), the challenge evolves. A representation that is merely sufficient for reconstruction but not minimal may still overfit in a more subtle way. It might encode spurious correlations or incidental stylistic details common across the dataset by chance, mistaking them for core domain principles. For example, it might learn that problem descriptions frequently use a specific turn of phrase, rather than capturing the abstract structural pattern of the domain itself. Therefore, our objective is to learn a minimal sufficient representation that is robust across this spectrum: it must be general enough to avoid overfitting on smaller datasets, yet discerning enough to separate the essential, generalizable domain characteristics from coincidental, sample-specific artifacts in larger ones. The following details DOMINO.

### 2.2 IMPLICIT DOMAIN REPRESENTATION LEARNING VIA PROMPT TUNING

To accomplish our goal of synthesizing domain-specific data, we need to (i) learn a suitable representation of the domain $\mathcal{D}$ and (ii) leverage that representation to generate new data within the domain. A promising approach is to employ a pre-trained LLM as an implicit encoder to represent the domain, capitalizing on its extensive knowledge base. Mathematically, let $\boldsymbol{D} = [\boldsymbol{d}_1, \cdots, \boldsymbol{d}_k]$ denote the domain representation comprising $k$ **domain soft tokens**, each with dimension $d$. We aim to optimize $\boldsymbol{D}$ such that $p(\boldsymbol{D}|\mathcal{X}) = \prod_{i=1}^{n} p(\boldsymbol{D}|\boldsymbol{X}^{(i)})$ is maximized. According to Bayes' rule,

$$p(\boldsymbol{D}|\boldsymbol{X}^{(i)}) = \frac{p(\boldsymbol{D})p(\boldsymbol{X}^{(i)}|\boldsymbol{D})}{p(\boldsymbol{X}^{(i)})} \propto p(\boldsymbol{X}^{(i)}|\boldsymbol{D}), \tag{1}$$

where $p(\boldsymbol{X}^{(i)}|\boldsymbol{D})$ signifies the probability of generating the data example $\boldsymbol{X}^{(i)}$ given the domain representation $\boldsymbol{D}$, as determined by an LLM. In this context, we assume a uniform prior distribution for the domain representation, $p(\boldsymbol{D}) \propto 1$, reflecting our lack of prior knowledge about the domain. Consequently, the problem amounts to soft prompt tuning, where our objective is to identify the domain-level soft token prompt $\boldsymbol{D}$ that maximizes the likelihood of all reference data $\boldsymbol{X}^{(1:n)}$:

$$\mathcal{L}_1 = -\frac{1}{n} \sum_{i=1}^{n} \log p(\boldsymbol{X}^{(i)}|\boldsymbol{D}). \tag{2}$$

Unlike explicit encoders that require structured inputs (e.g., labeled features), LLMs leverage their pre-training on vast corpora to infer domain-specific representations directly from natural language examples. Moreover, soft prompt tuning avoids full model fine-tuning, preserving the LLM's general capabilities while specializing its behavior for $\mathcal{D}$. In addition, the same framework applies to diverse domains without architectural changes, as the domain representation $\boldsymbol{D}$ adapts to each domain's unique characteristics. Subsequently, new data can be synthesized by inputting the estimated $\boldsymbol{D}$ into the LLM, drawing samples according to $p(\boldsymbol{X}|\boldsymbol{D})$, which will be discussed in Section 2.4.

## 2.3 MINIMAL SUFFICIENT DOMAIN REPRESENTATION LEARNING

While the soft prompt tuning approach described above seems promising, it suffers from a critical issue: overfitting. Given a small number of reference examples (tens to hundreds), the learned domain representation $\boldsymbol{D}$ can become overly tailored to the training data. This may result in synthetic data that closely resembles the reference examples, failing to capture the broader range of possibilities within the observation space $\mathcal{X}$ of the domain $\mathcal{D}$. Indeed, this approach primarily aims to learn a sufficient representation for the domain $\mathcal{D}$, which we define as follows:

**Definition 1.** *(Sufficiency) A representation $\boldsymbol{D}$ is considered sufficient for domain $\mathcal{D}$ if the conditional distribution $p(\boldsymbol{X}^{(i)}|\boldsymbol{D}, \mathcal{D}) = p(\boldsymbol{X}^{(i)}|\boldsymbol{D})$ for all $\boldsymbol{X}^{(i)} \in \mathcal{X}$. In other words, $I(\mathcal{D}; \boldsymbol{X}^{(i)}|\boldsymbol{D}) = 0$, indicating that $\boldsymbol{D}$ captures all information about $\mathcal{D}$ contained in $\boldsymbol{X}^{(i)}$, where $I$ is mutual information.*

However, sufficiency alone does not guarantee generalization. A representation that retains unnecessary details (e.g., unique attributes of individual examples) risks overfitting. To mitigate this, we seek a minimal sufficient representation $\boldsymbol{D}^*$:

**Definition 2.** *(Minimal Sufficiency) A sufficient representation $\boldsymbol{D}^*$ is minimal if and only if it is a function of every other sufficient representation $\boldsymbol{D}$, i.e., $\boldsymbol{D}^* = f(\boldsymbol{D})$. According to the deterministic function property of mutual information, $I(\boldsymbol{D}^*, \boldsymbol{X}^{(i)}) = I(f(\boldsymbol{D}), \boldsymbol{X}^{(i)}) \leq I(\boldsymbol{D}, \boldsymbol{X}^{(i)})$, indicating that $\boldsymbol{D}^*$ discards all information irrelevant to $\mathcal{D}$.*

Minimal sufficiency ensures $\boldsymbol{D}^*$ focuses on domain-wide patterns (e.g., the general topic, task type, or style) rather than memorizing sample-specific traits (e.g., specific facts, word choices, or constraints). This process is the computational analog to the inductive leap humans make: identifying the essential, shared "rules" of a concept while ignoring the incidental details of any single example. This enables the generation of diverse synthetic data that spans the observation space $\mathcal{X}$, rather than replicating training examples. To enforce minimal sufficiency, we explicitly separate domain-level and sample-level information by introducing sample-specific representations $\boldsymbol{S}^{(i)} = [\boldsymbol{s}_1^{(i)}, \cdots, \boldsymbol{s}_\ell^{(i)}]$, one for each individual example $\boldsymbol{X}^{(i)}$. These **sample-level soft tokens** encode unique aspects of each example $\boldsymbol{X}^{(i)}$. We optimize $\boldsymbol{D}^*$ and $\boldsymbol{S}^{(i)}$ via a contrastive loss:

$$\mathcal{L}_2 = -\frac{1}{n} \sum_{i=1}^{n} \log \frac{p(\boldsymbol{X}^{(i)}|\boldsymbol{D}^*, \boldsymbol{S}^{(i)})}{\sum_{j \neq i} p(\boldsymbol{X}^{(j)}|\boldsymbol{D}^*, \boldsymbol{S}^{(i)})}. \tag{3}$$

The numerator guarantees that $\boldsymbol{D}^*$ and $\boldsymbol{S}^{(i)}$ jointly reconstruct $\boldsymbol{X}^{(i)}$. On the other hand, the denominator penalizes $\boldsymbol{D}^*$ for encoding information that could facilitate $\boldsymbol{S}^{(i)}$ to reconstruct unrelated examples $\boldsymbol{X}^{(j)}$ for $j \neq i$. This design embodies the principle that $\boldsymbol{D}^*$ should capture shared aspects across all samples in the domain, while $\boldsymbol{S}^{(i)}$ focuses on the unique aspects of the specific sample $\boldsymbol{X}^{(i)}$. By forcing the LLM to rely on $\boldsymbol{S}^{(i)}$ to distinguish $\boldsymbol{X}^{(i)}$ from other samples $\boldsymbol{X}^{(j)}$, we prevent $\boldsymbol{D}^*$ from overfitting to the specifics of individual examples. This, in turn, promotes a more generalizable, minimal sufficient domain representation (i.e., an inductive abstraction). With this loss, we can prove:

**Proposition 1.** *Given $\boldsymbol{X}^{(i)} \in \boldsymbol{X}^{(1:n)}$, $\mathcal{L}_2$ directly maximizes $I(\boldsymbol{S}^{(i)}, \boldsymbol{X}^{(i)}|\boldsymbol{D}^*)$, ensuring that $\boldsymbol{S}^{(i)}$ captures unique information about $\boldsymbol{X}^{(i)}$.*

**Proposition 2.** *$\mathcal{L}_2$ directly minimizes $I(\boldsymbol{S}^{(i)}; \boldsymbol{D}^*)$, promoting disentanglement between $\boldsymbol{S}^{(i)}$ and $\boldsymbol{D}^*$*

The detailed proofs of **Proposition 1** and **Proposition 2** are listed in Appendix B.1 and B.2. Finally, the overall loss function for learning the minimal sufficient representation $\boldsymbol{D}^*$ for the domain $\mathcal{D}$ is a combination of the domain-level likelihood and the contrastive loss:

$$\mathcal{L} = \mathcal{L}_1 + \lambda \mathcal{L}_2, \tag{4}$$

where $\lambda$ is a weighting hyperparameter that controls the trade-off between maximizing data likelihood and promoting domain representation disentanglement. A higher $\lambda$ encourages the LLM to learn a more minimal, generalizable domain representation by preventing overfitting to individual samples. Further, we theoretically prove the connection between optimization objective $\mathcal{L}$ and the information-theoretic principles of minimal sufficiency (**Proposition 3**, proof in Appendix B.3).

**Proposition 3.** *The optimization objective $\mathcal{L}$ serves as a tractable proxy for learning a representation $\boldsymbol{D}^*$ that approximates the information-theoretic properties of a minimal sufficient statistic. Specifically, minimizing $\mathcal{L}$ jointly encourages: 1. **Sufficiency**: The maximization of mutual information $I(\boldsymbol{D}^*; \boldsymbol{X})$, ensuring $\boldsymbol{D}^*$ captures relevant domain information. 2. **Minimality**: The minimization of mutual information $I(\boldsymbol{D}^*; \boldsymbol{S})$, ensuring $\boldsymbol{D}^*$ is disentangled from sample-specific information.*

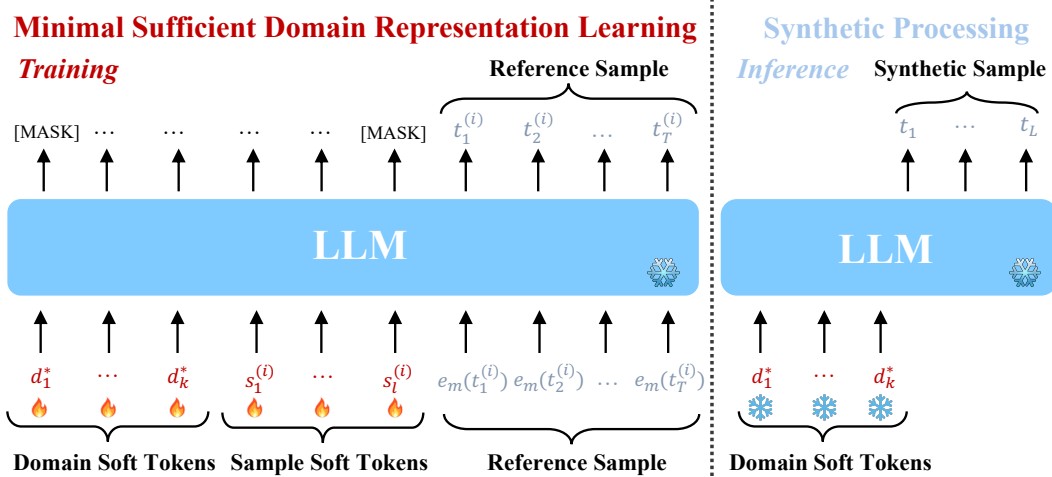

Figure 2: The left side represents the minimal sufficient domain representation learning process, where $t_1, \cdots, t_T$ denote the raw natural language tokens of a reference sample, $e_m$ denotes the corresponding embedding. Domain-level soft tokens $\boldsymbol{D}^*$ and sample-level soft tokens $\boldsymbol{S}^{(i)}$ are jointly optimized. The right side represents using only optimized $\boldsymbol{D}^*$ to synthesize domain-specific samples.

### 2.4 DATA SYNTHESIS

In practice, the minimal sufficient domain representation $\boldsymbol{D}^*$ and the sample-level prompt $\boldsymbol{S}^{(i)}$ are jointly optimized during training using the objective $\mathcal{L}$ in equation 4 (on the left side of Figure 2). Once training is complete, only the learned $\boldsymbol{D}^*$ is used to guide the LLM during the data synthesis phase (i.e., sampled from distribution $p(\boldsymbol{X}|\boldsymbol{D}^*)$), as shown on the right side of Figure 2. Notably, the LLM is fixed throughout both the training and synthesis stages. The key intuition is that by disentangling $\boldsymbol{D}^*$ from sample-specifics, we prevent it from merely memorizing the training examples and instead encourage the generation of more diverse outputs. **Proposition 4** provides the formal theoretical guarantee for this. It proves that, under quantifiable conditions, the distribution guided by our minimal sufficient representation, $p(\boldsymbol{X}|\boldsymbol{D}^*)$, has a strictly larger effective support (the set of high-probability outcomes) than the distribution $p_D$ guided by a standard, non-minimal prompt.

**Proposition 4.** *Let $p_D$ and $p_{D^*}$ denote the distributions $p(\boldsymbol{X}|\boldsymbol{D})$ (from vanilla prompt tuning) and $p(\boldsymbol{X}|\boldsymbol{D}^*)$ (from minimal sufficient, disentangled prompt tuning via $\mathcal{L}_1 + \lambda\mathcal{L}_2$), respectively. For any $\epsilon \in (0, 1/e)$, define the $\epsilon$-support of $p$, $\mathrm{supp}_\epsilon(p) = \{x : p(x) > \epsilon\}$, with cardinality $S_p^\epsilon = |\mathrm{supp}_\epsilon(p)|$. Define the uniformity gap for $p$ as $\delta_p = \log S_p^\epsilon - H(p)$, where $H(p)$ is the entropy of $p$. If*

$$H(p_{D^*}) - H(p_D) > \delta_{p_D} + \epsilon \log \frac{1}{\epsilon} \tag{5}$$

*then $S_{p_{D^*}}^\epsilon > S_{p_D}^\epsilon$; that is, the $\epsilon$-support of $p(\boldsymbol{X}|\boldsymbol{D}^*)$ is strictly larger than that of $p(\boldsymbol{X}|\boldsymbol{D})$.*

*Proof.* See Appendix B.4. $\square$

## 3 EXPERIMENTS

### 3.1 EXPERIMENTAL SETUP

**Domain and Benchmark Selection.** To effectively evaluate DOMINO, we require benchmarks that embody our core problem definition: domains where characteristics are implicit, evolving, and difficult to articulate textually. The recently proposed dynamic benchmark LiveCodeBench (Jain et al., 2024) provides a perfect testbed. In competitive programming, the "domain" is not a static category like "code generation", but the constantly evolving style, complexity, and set of algorithmic patterns characteristic of platforms during a specific time frame. It is notoriously difficult to write a single prompt that captures "the essence of a typical LeetCode problem from late 2024", yet it is easy to collect examples from that period. Following these principles, we select two distinct domains from the recently proposed dynamic benchmark LiveCodeBench (Jain et al., 2024): Live Code

Table 1: Main Results of DOMINO and the compared methods on the two target domains. DOMINO-Direct Domain is an ablation of our method using only $\mathcal{L}_1$ without $\mathcal{L}_2$.

| Methods | Live Code Generation | | | Live Code Execution |
|---|---|---|---|---|
| | Pass@1 | Pass@5 | Pass@10 | Pass@1 |
| Using OPENCODER-8B-Instruct Huang et al. (2024a) as Backbone | | | | |
| OPENCODER-8B-Base | 7.96 | 14.56 | 17.36 | 25.93 |
| OPENCODER-8B-Instruct | 10.00 | **17.23** | 19.76 | 37.96 |
| Reference SFT | 9.34 | 14.13 | 16.88 | 27.78 |
| MAGPIE-Human | 8.77 | 13.83 | 15.88 | 32.41 |
| MAGPIE-Few Shot | 9.81 | 15.32 | 17.49 | 34.26 |
| DOMINO-Direct Domain | 10.15 | 16.24 | 19.39 | 38.88 |
| DOMINO | **12.63** | 16.81 | **20.44** | **42.59** |
| Using QWEN2.5-CODER-7B-Instruct Hui et al. (2024) as Backbone | | | | |
| QWEN2.5-CODER-7B-Base | 9.04 | 15.35 | 17.96 | 45.37 |
| QWEN2.5-CODER-7B-Instruct | 16.88 | 21.73 | 23.35 | 55.55 |
| Reference SFT | 13.47 | 21.59 | 23.95 | 36.11 |
| MAGPIE-Human | 12.45 | 20.85 | 22.72 | 42.59 |
| MAGPIE-Few Shot | 14.36 | 20.98 | 24.23 | 48.15 |
| DOMINO-Direct Domain | 15.74 | 24.66 | 28.14 | 50.92 |
| DOMINO | **17.31** | **26.81** | **29.94** | **56.48** |
| Using QWEN2.5-CODER-14B-Instruct Hui et al. (2024) as Backbone | | | | |
| QWEN2.5-CODER-14B-Base | 19.46 | 26.42 | 27.4 | 45.37 |
| QWEN2.5-CODER-14B-Instruct | 23.35 | 30.13 | 31.73 | 59.26 |
| Reference SFT | 19.64 | 24.86 | 26.33 | 38.88 |
| MAGPIE-Human | 20.78 | 27.17 | 28.74 | 57.41 |
| MAGPIE-Few Shot | 20.12 | 25.27 | 27.45 | 57.41 |
| DOMINO-Direct Domain | 21.62 | 28.97 | 31.73 | 60.19 |
| DOMINO | **24.75** | **30.45** | **33.24** | **62.04** |

Generation and Live Code Execution. We define a temporal cutoff point for each domain: domain samples collected before this point are used as reference data (input-output pairs), while domain samples collected afterward serve as the test set. This setup directly tests the model's ability to induce generalizable principles rather than memorizing past examples. The details are in Appendix C.1.

**LLM Backbone Selection.** We select OPENCODER-7B (Huang et al., 2025a), QWEN2.5-CODER-7B (Hui et al., 2024), and QWEN2.5-CODER-14B (Hui et al., 2024) as the LLM backbones, as they are among the top-performing LLMs in general code tasks. We use these LLMs (*Instruction* version) to synthesize domain-specific data that aligns with the distributions of the two target domains.

**Baselines.** We detail the baseline methods; all corresponding prompts are provided in Appendix C.7.

- **Direct Prompt**: We directly prompt the LLM backbone (both the *Base* and *Instruction* versions) to assess the LLM backbone's basic capabilities on the two target domains.
- **Reference SFT**: We use the reference data from the target domain to perform supervised fine-tuning on the corresponding LLM *Base* version, in order to evaluate its domain adaptation capability.
- **MAGPIE-Human**: We manually inspect the reference data and attempt to summarize the domain characteristics in natural language, which are then used as the MAGPIE system prompt to guide the LLM in synthesizing domain-specific samples.
- **MAGPIE-Few Shot**: We randomly select three reference samples and prompt the LLM to summarize the underlying domain characteristics in natural language. This self-generated domain description is then used as the MAGPIE system prompt in synthesizing domain-specific samples.
- **DOMINO-Direct Domain**: We directly use the $\mathcal{L}_1$ objective in Section 2.2 to obtain the implicit domain representation, which is then used to guide the LLM in synthesizing domain samples.

**DOMINO Train Details.** For both DOMINO-Direct Domain and DOMINO, we use input sequences from domain reference samples for representation learning. We set the hyperparameter $\lambda$ to 1,

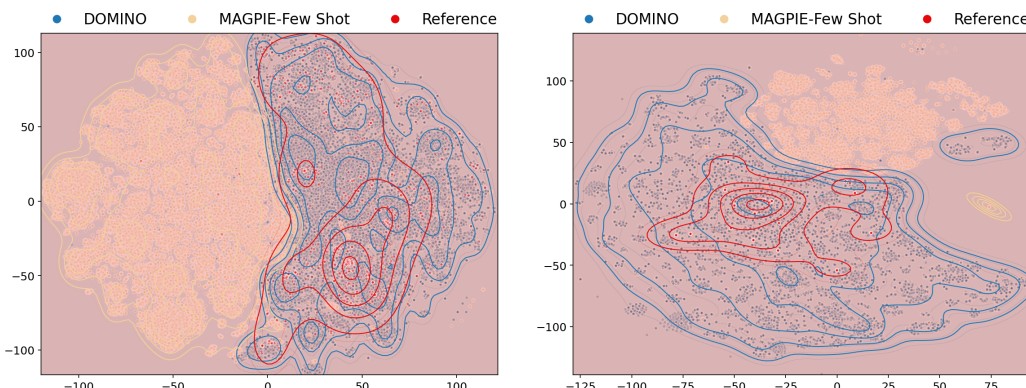

Figure 3: t-SNE of latent embeddings for synthetic and reference samples in *Live Code Generation* domain.

Figure 4: t-SNE of latent embeddings for synthetic and reference samples in *Live Code Execution* domain.

with 256 domain-level soft tokens in both settings, and 256 sample soft tokens additionally used in DOMINO. Note that the representation training is conducted using *Instruction* version of the LLM.

**Synthesis Process and Post-processing.** For MAGPIE-{Human, Few Shot}, DOMINO-Direct Domain, and DOMINO, we first synthesize domain-specific input sequences using vLLM (Kwon et al., 2023) with the *Instruction* version of LLMs, setting temperature $\mathcal{T}$ to 0.8. For fair comparison, we generate 80K input sequences for Live Code Generation domain and 40K for Live Code Execution domain across all four methods. A unified post-processing pipeline is then applied to filter out low-quality inputs (using the same LLM; details in Appendix C.8). After filtering, the same LLM is used to generate output sequences, forming input-output pairs for subsequent supervised fine-tuning.

**Domain Adaptation.** For all synthesized samples across methods generated using the *Instruction* version LLMs, we perform supervised fine-tuning on the corresponding *Base* versions using LLAMA-FACTORY (Zheng et al., 2024). Detailed training configurations are provided in Appendix C.9.

## 3.2 MAIN RESULTS

The main results are presented in Table 1, with each section block corresponding to one LLM backbone. We can find that: (i) Directly prompting the *Instruction* version of an LLM consistently outperforms its *Base* version across both domains, which is expected since instruction-tuned LLMs are explicitly aligned to follow system instructions. (ii) When fine-tuned directly on the domain reference data (Reference SFT), the LLM demonstrates a certain level of domain adaptation in the Live Code Generation domain. However, in the Live Code Execution domain, we observe a performance drop, suggesting that direct fine-tuning is prone to overfitting, especially in domains where the LLM already performs well out of the box. (iii) Both MAGPIE-Human and MAGPIE-Few Shot yield performance gains. Interestingly, MAGPIE-Few Shot slightly outperforms MAGPIE-Human, indicating that allowing the LLM to infer domain characteristics from a few reference samples is more effective than relying on manually written system prompts in unfamiliar domains. This highlights the LLM's capacity to capture subtle and latent domain patterns that may be overlooked by humans. (iv) Finally, DOMINO-Direct Domain and DOMINO achieve the best overall performance. Notably, DOMINO shows that synthetic domain-specific samples generated by the *Instruction* version of an LLM can be used to fine-tune its *Base* version, resulting in performance that even surpasses the original *Instruction* version. This finding suggests that the domain adaptation capabilities of LLMs can be further enhanced through self-generated, domain-aligned data.

## 3.3 DEEPER ANALYSIS

**Visualizing Synthetic Sample Distribution.** To better illustrate the relationship between synthesized and reference data distributions, we compare samples generated by DOMINO and MAGPIE-Few Shot. We use CODET5-EMBEDDING (Wang et al., 2023b) to obtain semantic embeddings and visualize them via t-SNE (van der Maaten & Hinton, 2008). As shown in Figure 3 and 4, DOMINO's synthesized samples are more widely dispersed in the embedding space and exhibit greater consistency with the distribution of reference samples. In contrast, the samples synthesized by MAGPIE-Few Shot are

Table 2: Qualitative Analysis: From Sample Mimicry to Domain Abstraction.

| Source | Example Problem |
|---|---|
| **Reference Sample** | **k-avoiding array**: Given `n` and `k`, find the minimum sum of a `k-avoiding` array of length `n` where no two distinct elements sum to `k`. |
| **Synthetic from $D$** (Overfitted) | **k-subsequence**: Given a string `n` of length `k`, find the number of `k-subsequences`. |
| **Synthetic from $D^*$** (Diverse but In-Domain) | 1. **Graph Path**: Find the distance between two airports in a tree-like network. 2. **Arithmetic Subsequence**: Find the longest arithmetic subsequence with a given `difference`. 3. **Palindrome Operations**: Find the minimum operations to make a string of digits a palindrome. |

Table 3: Results of DOMINO and the compared methods on the NLP tasks.

| Methods | IF Average | Instruct Following (IF) | | | |
|---|---|---|---|---|---|
| | | Paraphrase | Simplify | Story Gen. | Summarize |
| Qwen2.5-7B-Base | 44.50 | 41.23 | 48.27 | 53.84 | 34.65 |
| Qwen2.5-7B-Instruct | 51.91 | 50.62 | 60.83 | 54.58 | 41.61 |
| Reference SFT | 45.78 | 44.07 | 49.42 | 52.29 | 37.36 |
| MAGPIE Few Shot | 51.19 | 47.34 | 54.78 | 57.91 | 44.73 |
| DOMINO-Direct Domain | 53.78 | 52.49 | 58.37 | 57.91 | 46.38 |
| **DOMINO** | **55.39** | **54.28** | **61.96** | **58.23** | **47.12** |

cluster tightly and show weaker alignment with reference samples. These results highlight DOMINO's ability to generate diverse, domain-consistent data that better captures underlying domain traits.

**Interpretability of $D^*$.** To better interpret the specific domain patterns captured by $D^*$, we present a case study using samples that exemplify the clear distinction between the standard synthetic set ($D$) and the proposed diverse set ($D^*$). Critically, the synthetic samples from $D^*$ were specifically selected from regions of the t-SNE embedding space that are distant from the dense cluster representing $D$, visually confirming their diversity. The specific samples are provided in the Appendix C.2. The qualitative comparison (concluded in Table 2) provides concrete evidence: Even with a relatively large reference set (713 samples for Live Code Generation), the model trained with only a sufficiency objective ($D$, from $\mathcal{L}_1$) still overfits to superficial details. The addition of our contrastive disentanglement objective ($\mathcal{L}_2$) is crucial for pushing the model beyond mere memorization. It forces the model to learn a minimal representation ($D^*$) that successfully performs the inductive step of abstracting core domain principles, leading to the generation of truly diverse and novel samples.

**Generalization of DOMINO to More Task Domains.** To demonstrate DOMINO's effectiveness across a more diverse set of tasks, we have extended our experiments to the critical and well-defined *Instruction Following* **domain**. The detailed synthesis procedure and experimental results are provided in Appendix C.3 and Table 3. As shown, DOMINO consistently outperforms all baselines across the four subtasks, boosting the average score by 3.48 points over the strong instruction-tuned backbone and by 1.61 points over DOMINO-Direct Domain. These results reinforce the original findings and demonstrate that DOMINO's ability to learn and generalize from implicit domain characteristics is not limited to coding but extends to a diverse range of NLP tasks.

**Impact of Temperature $\mathcal{T}$.** In LLM-driven data synthesis, the temperature parameter controls the LLM's output diversity. To assess its impact on DOMINO, we vary $\mathcal{T}$ from 0.2 to 1.0. As shown in Table 4, DOMINO achieves relatively stable performance across different $\mathcal{T}$ settings. This robustness is attributed to the strong guidance of the domain representation $D^*$, which effectively constrains generation to preserve domain characteristics as much as possible.

**Impact of Domain-level Soft Token Counts.** In DOMINO, the domain-level soft tokens $[d_1^*, \cdots, d_k^*]$ are shared across all reference samples within the target domain and are designed to capture the domain's underlying characteristics. The number of domain soft tokens, $k$, is a tunable hyperparameter that determines the capacity of this representation. A larger $k$ enables the encoding of more domain-

Table 4: Results of varying $\mathcal{T}$ on synthetic process in the Live Code Generation using QWEN2.5-CODER-7B Instruction as the LLM backbone.

| Temperature | Live Code Generation | | |
|---|---|---|---|
| | Pass@1 | Pass@5 | Pass@10 |
| $\mathcal{T} = 0.2$ | 16.35 | 22.88 | 24.55 |
| $\mathcal{T} = 0.4$ | 16.77 | 24.67 | 27.54 |
| $\mathcal{T} = 0.6$ | 15.45 | 25.17 | 29.34 |
| $\mathcal{T} = 0.8$ | **17.31** | **26.81** | **29.94** |
| $\mathcal{T} = 1.0$ | 16.53 | 25.95 | 29.34 |

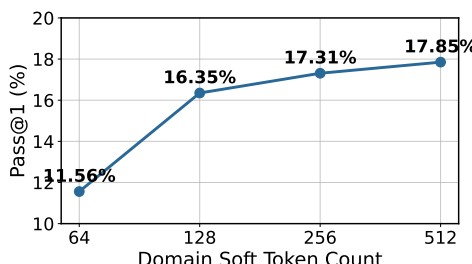

Figure 5: Results of the impact of domain soft token counts in Live Code Generation using QWEN2.5-CODER-7B Instruction as backbone.

specific features across a broader embedding space. To assess the impact of $k$, we vary it over the set $\{64, 128, 256, 512\}$. As shown in Figure 5, performance consistently improves with larger values of $k$, suggesting that higher-capacity domain representations better capture fine-grained domain characteristics and provide stronger guidance for synthetic data generation.

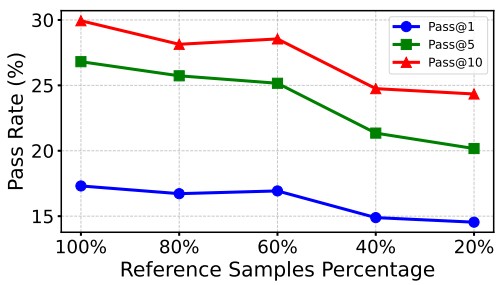

Figure 6: Impact of reference sample percentage in Live Code Generation domain using QWEN2.5-CODER-7B Instruction as the backbone.

Table 5: Results of varying $\lambda$ in DOMINO in the Live Code Generation domain using QWEN2.5-CODER-7B Instruction as LLM backbone.

| Hyperparameter $\lambda$ | Live Code Generation | | |
|---|---|---|---|
| | Pass@1 | Pass@5 | Pass@10 |
| $\lambda = 0.25$ | 15.38 | 22.44 | 24.38 |
| $\lambda = 0.5$ | 15.95 | 22.88 | 24.77 |
| $\lambda = 1$ | 17.31 | **26.81** | **29.94** |
| $\lambda = 2$ | **17.58** | 25.75 | 28.58 |
| $\lambda = 4$ | 16.58 | 26.38 | 28.89 |

**Impact of Reference Data Percentage.** In DOMINO, the domain characteristics are inserted into domain soft tokens by leveraging reference data. To evaluate the impact of reference data quantity, we vary the proportion of the original reference set from 100% to 20%. As shown in Figure 6, reducing the proportion of reference data consistently degrades performance. This aligns with the intuition: more reference samples provide a more comprehensive view of the domain, enabling more effective representation learning. Furthermore, we also evaluate the impact of reference data quantity on the Live Code Execution task; the corresponding results and analysis are presented in Appendix C.4.

**Impact of Hyperparameter $\lambda$.** In the original DOMINO optimization objective, the hyperparameter $\lambda$ controls the trade-off between maximizing data likelihood and promoting domain representation disentanglement. To evaluate its impact on performance, we vary $\lambda$ across the set $\{0.25, 0.5, 1, 2, 4\}$. As shown in Table 5, setting $\lambda \geq 1$ leads to a noticeable performance improvement compared to $\lambda < 1$. This indicates that assigning greater weight to disentanglement helps the LLM learn a more minimal and generalizable domain representation by mitigating overfitting to individual samples.

**Further Analysis.** 1. **Case studies** provide an intuitive representation of domain-specific synthetic samples in Appendix C.10; 2. A comparison of **synthetic against reference samples, using equal numbers**, shown in Appendix C.5; 3. **More samples for MAGPIE-Few Shot** in Appendix C.6.

## 4 CONCLUSION

In this work, we propose DOMINO, a method for domain-specific data synthesis under implicit supervision. By learning minimal sufficient domain representations from reference data, DOMINO enables scalable domain-specific data synthesis without manual prompt engineering. Theoretical and empirical results show that DOMINO can capture essential characteristics of the domain, generate diverse, domain-aligned samples even without explicit domain definitions or any prior knowledge.

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

## A  RELATED WORK

In recent years, with the advancement of LLMs, data synthesis has attracted increasing research attention due to its unique advantages (Li et al., 2024; Long et al., 2024; Huang et al., 2025b). Broadly, existing LLM-driven synthetic data approaches can be categorized into three main categories:

**Instruction Evolution:** This category involves leveraging iterative and carefully crafted prompt engineering to guide LLMs in expanding instruction sets. Specifically, Self-Instruct (Wang et al., 2023a) employs a curated seed pool and task-specific prompts to generate additional instruction data. Meanwhile, Evol-Instruct (Xu et al., 2024) uses strategically designed prompts to guide LLMs in modifying existing instructions or creating new ones from both depth and breadth perspectives. While effective, these methods rely heavily on complex natural prompt engineering within the domain.

**Key-Point-Driven:** These methods guide LLMs to synthesize data by extracting knowledge and key points from the target domain. Li et al. (2024) employ GPT-4 to construct a taxonomy of concepts; however, the resulting synthetic data distribution deviates significantly from real data. Huang et al. (2024b) build an explicit concept pool from extracted domain concepts. In contrast, KPDDS (Huang et al., 2024a) constructs topic–key point pairs from a large number of seed examples to capture the frequency of domain-specific topics. It then samples multiple topic–key point combinations to guide the generation of new samples. Despite their effectiveness, these approaches rely heavily on extensive prior knowledge and assume that the target domain possesses a hierarchical conceptual structure. Moreover, they require such structures and key points can be explicitly described in natural language.

**System Prompt Guided:** Recently, MAGPIE (Xu et al., 2025) introduced a method that constructs instruction data by prompting aligned LLMs with a pre-defined query template. By incorporating a domain-specific system prompt, MAGPIE can synthesize large-scale domain data. However, it necessitates the deliberate design of the system prompt to guide the LLM toward the target domain.

However, all three categories of methods break down in real-world scenarios when the target domain cannot be explicitly described in natural language and no prior domain knowledge is available. In contrast, our proposed method, DOMINO, effectively overcomes this limitation by enabling the synthesis of a virtually unlimited quantity of target-domain data without requiring any prior knowledge or manual prompt engineering.

## B THEORETICAL GUARANTEE

### B.1 THE PROOF OF PROPOSITION 1.

***Proposition***: Given $\boldsymbol{X}^{(i)} \in \boldsymbol{X}^{(1:n)}$, $\mathcal{L}_2$ directly maximizes $I(\boldsymbol{S}^{(i)}, \boldsymbol{X}^{(i)}|\boldsymbol{D}^*)$, ensuring that $\boldsymbol{S}^{(i)}$ captures unique information about $\boldsymbol{X}^{(i)}$.

*Proof.* By definition:

$$I(\boldsymbol{S}^{(i)}; \boldsymbol{X}^{(i)}|\boldsymbol{D}^*) = \mathbb{E}_{p(\boldsymbol{D}^*, \boldsymbol{S}^{(i)}, \boldsymbol{X}^{(i)})}\left[\log \frac{p(\boldsymbol{S}^{(i)}|\boldsymbol{D}^*, \boldsymbol{X}^{(i)})}{p(\boldsymbol{S}^{(i)}|\boldsymbol{D}^*)}\right], \tag{6}$$

where $\mathbb{E}_{p(\boldsymbol{D}^*, \boldsymbol{S}^{(i)}, \boldsymbol{X}^{(i)})}$ represents the expectation over the distribution $p(\boldsymbol{D}^*, \boldsymbol{S}^{(i)}, \boldsymbol{X}^{(i)})$. Applying Bayes' theorem, we can express the numerator as:

$$p(\boldsymbol{S}^{(i)}|\boldsymbol{D}^*, \boldsymbol{X}^{(i)}) = \frac{p(\boldsymbol{X}^{(i)}|\boldsymbol{D}^*, \boldsymbol{S}^{(i)})p(\boldsymbol{S}^{(i)}|\boldsymbol{D}^*)}{p(\boldsymbol{X}^{(i)}|\boldsymbol{D}^*)}. \tag{7}$$

On the other hand, the denominator can be expanded as:

$$p(\boldsymbol{S}^{(i)}|\boldsymbol{D}^*) = \sum_{j=1}^{n} p(\boldsymbol{X}^{(j)}, \boldsymbol{S}^{(i)}|\boldsymbol{D}^*) = \sum_{j=1}^{n} p(\boldsymbol{X}^{(j)}|\boldsymbol{D}^*, \boldsymbol{S}^{(i)})p(\boldsymbol{S}^{(i)}|\boldsymbol{D}^*). \tag{8}$$

$$
\begin{aligned}
I(\boldsymbol{S}^{(i)}; \boldsymbol{X}^{(i)}|\boldsymbol{D}^*) &= \mathbb{E}_{p(\boldsymbol{D}^*, \boldsymbol{S}^{(i)}, \boldsymbol{X}^{(i)})}\left[\log \frac{p(\boldsymbol{X}^{(i)}|\boldsymbol{D}^*, \boldsymbol{S}^{(i)})}{p(\boldsymbol{X}^{(i)}|\boldsymbol{D}^*)\sum_{j=1}^{n} p(\boldsymbol{X}^{(j)}|\boldsymbol{D}^*, \boldsymbol{S}^{(i)})}\right], \\
&\geq \mathbb{E}_{p(\boldsymbol{D}^*, \boldsymbol{S}^{(i)}, \boldsymbol{X}^{(i)})}\left[\log \frac{p(\boldsymbol{X}^{(i)}|\boldsymbol{D}^*, \boldsymbol{S}^{(i)})}{\sum_{j=1}^{n} p(\boldsymbol{X}^{(j)}|\boldsymbol{D}^*, \boldsymbol{S}^{(i)})}\right], \\
&= \mathbb{E}_{p(\boldsymbol{D}^*, \boldsymbol{S}^{(i)}, \boldsymbol{X}^{(i)})}\left[\log \frac{p(\boldsymbol{X}^{(i)}|\boldsymbol{D}^*, \boldsymbol{S}^{(i)})}{p(\boldsymbol{X}^{(i)}|\boldsymbol{D}^*, \boldsymbol{S}^{(i)}) + \sum_{j\neq i} p(\boldsymbol{X}^{(j)}|\boldsymbol{D}^*, \boldsymbol{S}^{(i)})}\right], \\
&\geq -\mathcal{L}_2
\end{aligned}
\tag{9}
$$

The inequality holds because $p(\boldsymbol{X}^{(i)}|\boldsymbol{D}^*)$ is a categorical distribution (i.e. $p(\boldsymbol{X}^{(i)}|\boldsymbol{D}^*) \leq 1$) and $p(\boldsymbol{X}^{(i)}|\boldsymbol{D}^*, \boldsymbol{S}^{(i)})$ is always non-negative. So, minimizing $\mathcal{L}_2$ directly maximizes $I(\boldsymbol{S}^{(i)}, \boldsymbol{X}^{(i)}|\boldsymbol{D}^*)$. $\square$

### B.2 THE PROOF OF PROPOSITION 2.

***Proposition***: $\mathcal{L}_2$ directly minimizes $I(\boldsymbol{S}^{(i)}; \boldsymbol{D}^*)$, promoting the disentanglement between $\boldsymbol{S}^{(i)}$ and $\boldsymbol{D}^*$.

*Proof.* Given the limited capacity (or size) of $\boldsymbol{S}^{(i)}$, the total mutual information $I(\boldsymbol{S}^{(i)}; \boldsymbol{X}^{(i)})$ is bounded. In practice, the combined length of the domain-level prompt $\boldsymbol{D}^*$ and the sample-level prompt $\boldsymbol{S}^{(i)}$ is generally smaller than the length of the observed data $\boldsymbol{X}^{(i)}$, thus, this assumption is valid. By applying the chain rule of mutual information, we have:

$$I(\boldsymbol{S}^{(i)}; \boldsymbol{X}^{(i)}) = I(\boldsymbol{S}^{(i)}; \boldsymbol{D}^*) + I(\boldsymbol{S}^{(i)}; \boldsymbol{X}^{(i)}|\boldsymbol{D}^*). \tag{10}$$

Since $\mathcal{L}_2$ maximizes $I(\boldsymbol{S}^{(i)}; \boldsymbol{X}^{(i)}|\boldsymbol{D}^*)$ as proven in **Proposition** 1 and the total $I(\boldsymbol{S}^{(i)}; \boldsymbol{X}^{(i)})$ is bounded, it necessitates the minimization of $I(\boldsymbol{S}^{(i)}; \boldsymbol{D}^*)$ to maintain the equation's integrity. $\square$

## B.3 THE PROOF OF PROPOSITION 3.

**Proposition**: The optimization objective $\mathcal{L} = \mathcal{L}_1 + \lambda\mathcal{L}_2$ serves as a tractable proxy for learning a representation $\boldsymbol{D}^*$ that approximates the information-theoretic properties of a minimal sufficient statistic. Specifically, minimizing $\mathcal{L}$ jointly encourages:

1. **Sufficiency**: The maximization of mutual information $I(\boldsymbol{D}^*; \boldsymbol{X})$, ensuring $\boldsymbol{D}^*$ captures relevant domain information.

2. **Minimality**: The minimization of mutual information $I(\boldsymbol{D}^*; \boldsymbol{S})$, ensuring $\boldsymbol{D}^*$ is disentangled from sample-specific information.

*Proof.* The proof analyzes the two components of the loss function and assume that the empirical loss, calculated over the reference samples, is a reasonable estimator of the expected loss over the true data distribution.

**Part 1: Minimizing $\mathcal{L}_1$ Encourages Sufficiency.**

The first loss term, $\mathcal{L}_1$, is the negative log-likelihood of the data $\boldsymbol{X}$ given the domain representation $\boldsymbol{D}^*$. In expectation, minimizing this empirical loss corresponds to minimizing the conditional entropy $H(\boldsymbol{X}|\boldsymbol{D}^*)$:

$$\min(\mathcal{L}_1) \implies \min H(\boldsymbol{X}|\boldsymbol{D}^*) \tag{11}$$

The mutual information $I(\boldsymbol{D}^*; \boldsymbol{X})$ is defined as $I(\boldsymbol{D}^*; \boldsymbol{X}) = H(\boldsymbol{X}) - H(\boldsymbol{X}|\boldsymbol{D}^*)$. Since the data entropy $H(\boldsymbol{X})$ is a constant with respect to the model parameters, minimizing the conditional entropy $H(\boldsymbol{X}|\boldsymbol{D}^*)$ is equivalent to maximizing the mutual information $I(\boldsymbol{D}^*; \boldsymbol{X})$.

$$\min H(\boldsymbol{X}|\boldsymbol{D}^*) \iff \max I(\boldsymbol{D}^*; \boldsymbol{X}) \tag{12}$$

Thus, minimizing $\mathcal{L}_1$ drives the representation $\boldsymbol{D}^*$ to capture as much information as possible about the data $\boldsymbol{X}$, fulfilling the sufficiency criterion.

**Part 2: Minimizing $\mathcal{L}_2$ Encourages Minimality.**

The second loss term, $\mathcal{L}_2$, is a contrastive loss. As established in **Proposition** 1, minimizing $\mathcal{L}_2$ maximizes the conditional mutual information $I(\boldsymbol{S}; \boldsymbol{X}|\boldsymbol{D}^*)$. To understand how this promotes minimality, we use the chain rule of mutual information:

$$I(\boldsymbol{S}; \boldsymbol{D}^*) = I(\boldsymbol{S}; \boldsymbol{X}) - I(\boldsymbol{S}; \boldsymbol{X}|\boldsymbol{D}^*) \tag{13}$$

The sample-specific representation $\boldsymbol{S}$ has a fixed information capacity, bounded by its entropy $H(\boldsymbol{S})$, which is determined by the model's architecture. This means $I(\boldsymbol{S}; \boldsymbol{X}) \leq H(\boldsymbol{S})$. This fixed capacity creates an "information budget" for what $\boldsymbol{S}$ can learn about $\boldsymbol{X}$. This budget is partitioned between information that is also in $\boldsymbol{D}^*$ (the shared information, $I(\boldsymbol{S}; \boldsymbol{D}^*)$) and information that is unique to $\boldsymbol{S}$ given $\boldsymbol{D}^*$ (the unique information, $I(\boldsymbol{S}; \boldsymbol{X}|\boldsymbol{D}^*)$). Since the total budget $I(\boldsymbol{S}; \boldsymbol{X})$ is bounded by the constant $H(\boldsymbol{S})$, and the objective of minimizing $\mathcal{L}_2$ is to maximize $I(\boldsymbol{S}; \boldsymbol{X}|\boldsymbol{D}^*)$, the optimization is strongly incentivized to allocate as much of the budget as possible to $I(\boldsymbol{S}; \boldsymbol{X}|\boldsymbol{D}^*)$. This necessarily encourages the minimization of $I(\boldsymbol{S}; \boldsymbol{D}^*)$. By the symmetry of mutual information, minimizing $I(\boldsymbol{S}; \boldsymbol{D}^*)$ is equivalent to minimizing $I(\boldsymbol{D}^*; \boldsymbol{S})$. This drives the domain representation $\boldsymbol{D}^*$ to be uninformative about the sample-specific details $\boldsymbol{S}$, thus satisfying the **minimality** criterion. $\qquad\square$

## B.4 THE PROOF OF PROPOSITION 4.

**Proposition**: Let $p_D$ and $p_{D^*}$ denote the distributions $p(\boldsymbol{X}|\boldsymbol{D})$ (from vanilla prompt tuning) and $p(\boldsymbol{X}|\boldsymbol{D}^*)$ (from minimal sufficient, disentangled prompt tuning via $\mathcal{L}_1 + \lambda\mathcal{L}_2$), respectively. For any $\epsilon \in (0, 1/e)$, define the $\epsilon$-support of $p$, $\mathrm{supp}_\epsilon(p) = \{x : p(x) > \epsilon\}$, with cardinality $S_p^\epsilon = |\mathrm{supp}_\epsilon(p)|$. Define the uniformity gap for $p$ as $\delta_p = \log S_p^\epsilon - H(p)$, where $H(p)$ is the entropy of $p$. If

$$H(p_{D^*}) - H(p_D) > \delta_{p_D} + \epsilon \log \frac{1}{\epsilon} \tag{14}$$

then $S_{p_{D^*}}^\epsilon > S_{p_D}^\epsilon$; that is, the $\epsilon$-support of $p(\boldsymbol{X}|\boldsymbol{D}^*)$ is strictly larger than that of $p(\boldsymbol{X}|\boldsymbol{D})$.

*Proof.* By definition, $\log S_p^\epsilon = H(p) + \delta_p$ and, by a standard entropy-support bound, for any $p$, $H(p) \leq \log S_p^\epsilon + \epsilon \log \frac{1}{\epsilon} \implies -\delta_p \leq \epsilon \log \frac{1}{\epsilon}$. Thus,

$$\log S_{p_{D*}}^\epsilon - \log S_{p_D}^\epsilon = [H(p_{D*}) - H(p_D)] + [\delta_{p_{D*}} - \delta_{p_D}] \tag{15}$$

$$> [\delta_{p_D} + \epsilon \log \frac{1}{\epsilon}] + [\delta_{p_{D*}} - \delta_{p_D}] \tag{16}$$

$$= \delta_{p_{D*}} + \epsilon \log \frac{1}{\epsilon} > -\epsilon \log \frac{1}{\epsilon} + \epsilon \log \frac{1}{\epsilon} = 0, \tag{17}$$

since,

$$-\delta_{p_{D*}} \leq \epsilon \log \frac{1}{\epsilon} \implies \delta_{p_{D*}} \geq -\epsilon \log \frac{1}{\epsilon} \tag{18}$$

$$\tag{19}$$

Therefore,

$$\log S_{p_{D*}}^\epsilon > \log S_{p_D}^\epsilon, i.e., S_{p_{D*}}^\epsilon > S_{p_D}^\epsilon \tag{20}$$

This closes the proof. □

## C  EXPERIMENTAL DETAILS

### C.1  TARGET DOMAIN DATA STATISTICS.

We select two distinct domains from the recently proposed dynamic benchmark LiveCodeBench (Jain et al., 2024): Live Code Generation and Live Code Execution. To ensure a fair and realistic evaluation that mimics a real-world "live" setting, we apply a consistent temporal cutoff: specifically, we use all domain samples released before the final update's timestamp as reference data (input–output pairs), while every sample that appears in that final update constitutes the test set.

For the **Live Code Generation** domain, the data collected before *September 2024* as reference data and data collected after that as the test set.

For the **Live Code Execution** domain, the data collected before *October 2023* as reference data (after deduplication) and data collected after that as the test set.

Table 6:  Statistics of the two target domains: Live Code Generation and Live Code Execution.

| | Reference Samples | Test Set | Temporal Cutoff Point |
|---|---|---|---|
| **Live Code Generation** | 713 | 167 | September 2024 |
| **Live Code Execution** | 66 | 108 | October 2023 |

**Target Domain Instances**. We present detailed examples of the reference data for the two target domains in Figure 10 and Figure 11.

**Evaluation:** For evaluation, we use the official evaluation scripts from LiveCodeBench and report the corresponding Pass@k metrics. The inference prompts used for the two target domains are shown in Figure 7 and Figure 8.

### C.2  CASE STUDY: INTERPRETABILITY OF $D^*$.

To better interpret the specific domain patterns captured by $D^*$, we present a case study using samples that exemplify the clear distinction between the standard synthetic set ($D$) and our proposed diverse set ($D^*$). Critically, the synthetic samples from $D^*$ are specifically selected from regions of the t-SNE embedding space that are distant from the dense cluster representing $D$, visually confirming their diversity. The specific samples, provided in Figure 9, reveal two key findings:

- **Overfitting in $D$**: The sample from $D$ demonstrates **sample-level mimicry**. It not only adopts the high-level structure (problem statement $\rightarrow$ examples $\rightarrow$ constraints) but also copies superficial

```
You are an expert Python programmer. You will be given a question
↪  (problem specification) and will generate a correct Python
↪  program that matches the specification and passes all tests.
↪  You will NOT return anything except for the program

### Question:\n{question.question_content}

{ if question.starter_code }
### Format: {PromptConstants.FORMATTING_MESSAGE}

```python
{question.starter_code}
```
{ else }
### Format: {PromptConstants.FORMATTING_WITHOUT_STARTER_MESSAGE}

```python
# YOUR CODE HERE
```
{ endif }

### Answer: (use the provided format with backticks)
```

Figure 7: Inference Prompt for *Live Code Generation* domain.

details like variable names (n, k) and problem naming conventions (k-xxx), indicating it has overfitted to specific instances.

- **Domain Abstraction in $D^*$** : In contrast, samples from $D^*$ capture the abstract **domain structure** of a competitive programming problem while introducing genuine topical diversity. They explore entirely new problem types—spanning graph theory, dynamic programming, and string algorithms—that are distinct from the reference sample.

## C.3 GENERALIZATION OF DOMINO TO MORE TASK DOMAINS.

To demonstrate DOMINO's effectiveness across a more diverse set of tasks, we extended our experiments to the critical and well-defined *Instruction Following* **domain**. We chose this domain for two key reasons:

- **Diverse Subtasks**: It comprises a variety of core language tasks (paraphrasing, summarization, simplification, story generation), allowing us to test DOMINO's versatility.
- **Challenging, Unseen Data**: We use LiveBench (White et al., 2025), a dynamic benchmark whose data is continuously updated. This ensures our test set is free from contamination and that even SOTA models do not achieve perfect scores, providing a meaningful test of generalization. Proving that DOMINO's synthetic data can improve performance on such a challenging, live benchmark is a strong testament to its value.

Following the protocol from main experiment of Live Code Generation and Live Code Execution, we use the temporal cutoff *(June 30, 2024)* to create reference and test sets. The reference set and test set each contain 200 samples. We then use DOMINO with a Qwen2.5-7B-Instruct (Team, 2024) backbone to synthesize 40K samples for fine-tuning. The results, evaluated using the official LiveBench suite, are presented in Table 3. As shown, DOMINO consistently outperforms all baselines across the four subtasks, boosting the average score by 3.48 points over the strong instruction-tuned backbone and by 1.61 points over the standard prompt-tuning approach (DOMINO-Direct Domain). These results reinforce the original findings and demonstrate that DOMINO's ability to learn and generalize from implicit domain characteristics is not limited to coding but extends to a diverse range of NLP tasks.

```
You are given a Python function and an assertion containing an
↪  input to the function. Complete the assertion with a literal
↪  (no unsimplified expressions, no function calls) containing the
↪  output when executing the provided code on the given input,
↪  even if the function is incorrect or incomplete. Do NOT output
↪  any extra information. Execute the program step by step before
↪  arriving at an answer, and provide the full assertion with the
↪  correct output in [ANSWER] and [/ANSWER] tags, following the
↪  examples.

[PYTHON]
def performOperation(s):
    s = s + s
    return "b" + s + "a"
assert performOperation(s = "hi") == ??
[/PYTHON]
[THOUGHT]
Let's execute the code step by step:

1. The function performOperation is defined, which takes a single
↪  argument s.
2. The function is called with the argument "hi", so within the
↪  function, s is initially "hi".
3. Inside the function, s is concatenated with itself, so s becomes
↪  "hihi".
4. The function then returns a new string that starts with "b",
↪  followed by the value of s (which is now "hihi"), and ends with
↪  "a".
5. The return value of the function is therefore "bhihia".
[/THOUGHT]
[ANSWER]
assert performOperation(s = "hi") == "bhihia"
[/ANSWER]

[PYTHON]
{code}
assert {input} == ??
[/PYTHON]
[THOUGHT]
```

Figure 8: Inference Prompt for *Live Code Execution* domain.

## C.4 IMPACT OF REFERENCE DATA PERCENTAGE ON LIVE CODE EXECUTION.

To evaluate the impact of reference data quantity on the Live Code Execution domain, we mirror the methodology of Figure 6 (Live Code Generation) using the QWEN2.5-CODER-7B Instruct backbone.

We vary the proportion of the reference set from 100% down to 20% and the results are shown in Figure 12. Figure 12 confirms the conclusion: reducing the amount of in-domain reference data leads to a corresponding drop in performance. The results also reveal a performance plateau, with almost no improvement from 80% to 100% of the reference data. This reinforces the central claim by suggesting that while a sufficient quantity of reference data is crucial for high performance, gains diminish once a point of saturation is reached. Determining this optimal quantity across different domains remains a compelling question for future work.

## C.5 COMPARISON OF SYNTHETIC SAMPLES VS. REFERENCE SAMPLES (EQUAL SIZE).

To further compare the performance gap when using the same number of DOMINO's synthetic samples versus original reference samples, we conducted an SFT experiment comparing the two.

**Reference Sample**

You are given two integers, n and k. An array of distinct positive integers is called a k-avoiding array if there does not exist any pair of distinct elements that sum to k. Return the minimum possible sum of a k-avoiding array of length n.

Example 1:
Input: n = 5, k = 4
Output: 18
Explanation: Consider the k-avoiding array [1,2,4,5,6], which has a sum of 18. It can be proven that there is no k-avoiding array with a sum less than 18.

Constraints:
1 <= n, k <= 50

**Synthetic Samples From D**

You are given a string n of length k. A string is called k-subsequence of n if n can be written as t1t2...tk, where every ti is a non-empty substring of n. Return the number of k-subsequences of n, given that k is a divisor of n. Since the answer may be very large, return the answer modulo 109 + 7. A substring is a contiguous sequence of characters within a string.

Example 1:
Input: n = "ababa", k = 3
Output: 7
Explanation:
The 7 3-subsequences of n are: "aba", "aba", "baba", "aba", "baba", "aba", "aba". Note that some subsequences are repeated, but we only need to count distinct subsequences.

Constraints:
1 <= n.length <= 10^4
n consists of lowercase English letters.
1 <= k <= 10

**Synthetic Samples From D***

You are given an integer array arr and an integer difference. Return the length of the longest subsequence in arr which is an arithmetic sequence such that the difference between adjacent elements in the subsequence is difference. A subsequence is a sequence that can be derived from another sequence by deleting some or no elements without changing the order of the remaining elements.

Example 1:
Input: arr = [1,2,3,4], difference = 1
Output: 4
Explanation: The longest arithmetic subsequence is [1,2,3,4].

Constraints:
1 <= arr.length <= 10^5
-10^4 <= arr[i],
difference <= 10^4

Figure 9: Synthetic Samples from $D$ and $D^*$.

**Reference Samples in *Live Code Generation* Domain.**

You are given two integers, n and k. An array of distinct positive integers is called a k-avoiding array if there does not exist any pair of distinct elements that sum to k. Return the minimum possible sum of a k-avoiding array of length n.

Example 1:
Input: n = 5, k = 4
Output: 18
Explanation: Consider the k-avoiding array [1,2,4,5,6], which has a sum of 18. It can be proven that there is no k-avoiding array with a sum less than 18.

Example 2:
Input: n = 2, k = 6
Output: 3
Explanation: We can construct the array [1,2], which has a sum of 3. It can be proven that there is no k-avoiding array with a sum less than 3.

Constraints:
1 <= n, k <= 50

Given a positive integer num represented as a string, return the integer num without trailing zeros as a string.

Example 1:

Input: num = "51230100"
Output: "512301"
Explanation: Integer "51230100" has 2 trailing zeros, we remove them and return integer "512301".

Example 2:

Input: num = "123"
Output: "123"
Explanation: Integer "123" has no trailing zeros, we return integer "123".

Constraints:

1 <= num.length <= 1000
num consists of only digits.
num doesn't have any leading zeros.

You are given a 0-indexed integer array nums. You have to find the maximum sum of a pair of numbers from nums such that the maximum digit in both numbers are equal. Return the maximum sum or -1 if no such pair exists.

Example 1:

Input: nums = [51,71,17,24,42]
Output: 88
Explanation:
For i = 1 and j = 2, nums[i] and nums[j] have equal maximum digits with a pair sum of 71 + 17 = 88.
For i = 3 and j = 4, nums[i] and nums[j] have equal maximum digits with a pair sum of 24 + 42 = 66.
It can be shown that there are no other pairs with equal maximum digits, so the answer is 88.
Example 2:

Input: nums = [1,2,3,4]
Output: -1
Explanation: No pair exists in nums with equal maximum digits.

Constraints:

2 <= nums.length <= 100
1 <= nums[i] <= 10^4

Figure 10: Detailed examples of the reference data in *Live Code Generation* domain.

**Reference Samples in *Live Code Execution* Domain.**

```python
def smallestString(s: str) -> str:
    if s == "a" * len(s):
        return "a"*(len(s) - 1)+"z"

    r = ""
    p = 0

    for i in s:
        if p == 1:
            if i == "a":
                p = 2
                r += i
            else:
                r += chr(ord(i) - 1)
        elif p == 0:
            if i == "a":
                r += i
            else:
                p = 1
                r += chr(ord(i) - 1)
        else:
            r += i

    return r

assert smallestString(s = 'acbbc')
== ?
```

```python
def minimumBeautifulSubstrings(s: str) ->
int:
    good = []
    num = 1
    n = len(s)
    while True:
        b = bin(num)[2:]
        if len(b) > n:
            break
        good.append(b)
        num *= 5
    dp = [int(1e9)] * (n + 1)
    dp[0] = 0
    gs = set(good)
    for i in range(n):
        for j in range(i + 1):
            if s[j:i + 1] in gs:
                dp[i + 1] = min(dp[i + 1],
dp[j] + 1)
    return -1 if dp[n] == int(1e9) else
dp[n]

assert minimumBeautifulSubstrings(s = '0')
== ?
```

Figure 11: Detailed examples of the reference data in *Live Code Execution* domain.

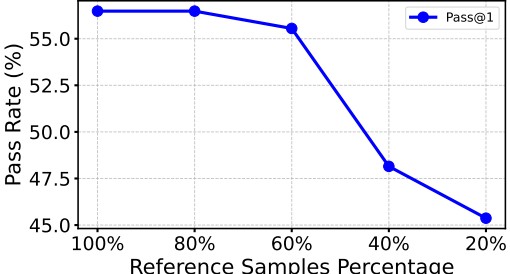

Figure 12: Result of the impact of reference sample percentage in Live Code Execution domain using QWEN2.5-CODER-7B Instruction as backbone.

Table 7: Performance Comparison of Reference vs. DOMINO's Synthetic Samples (Equal Size).

| Task | Metric | Reference Samples | DOMINO's Synthetic Samples |
|------|--------|-------------------|----------------------------|
| **Live Code Generation** | Pass@1 | 13.47 | 12.69 |
| | Pass@5 | 21.59 | 20.73 |
| | Pass@10 | 23.95 | 23.35 |
| **Live Code Execution** | Pass@1 | 36.11 | 34.26 |

```
### Characteristics of the current coding task:

1. The python function involves array manipulation and logical
↪  operations. The functions typically take arrays or lists as
↪  input and perform operations to derive or validate certain
↪  properties of these arrays.

2. The function input also showcase the use of assertions to
↪  validate the correctness of the functions.
```

Figure 13: MAGPIE-Few Shot for for *Live Code Execution* domain.

As shown in Table 7, the DOMINO's synthetic data achieves performance very close to the original data. This small gap is expected, as the original reference samples represent the ground-truth "gold" data for the domain. This result validates the high per-sample quality of DOMINO's generated sample.

## C.6 IMPACT OF MORE REFERENCE SAMPLES FOR MAGPIE.

In our comparative method, MAGPIE-Few shot, we originally randomly selected three reference samples to include in the prompt for domain representation summarization using the LLM. This setup considered that each reference sample typically contains several hundred tokens. To further investigate whether providing more reference samples to MAGPIE method would lead to performance gains, we conducte the corresponding experiment, with results presented in Table 8.

Table 8: Results of more reference samples for MAGPIE-Few shot.

| Live Code Generation | Pass@1 | Pass@5 | Pass@10 |
|----------------------|--------|--------|---------|
| MAGPIE Few-Shot = 3 | 14.36 | 20.98 | 24.23 |
| MAGPIE Few-Shot = 5 | 13.89 | 21.21 | 23.35 |
| MAGPIE Few-Shot = 10 | 11.81 | 16.82 | 18.39 |

As shown in Table 8, the MAGPIE performance remains comparable when using 3 or 5 few-shot samples. However, increasing the count to 10 severely challenges the LLM's ability to extract domain-specific information from the extended context. Our empirical evidence indicates that this larger few-shot count actually compromises performance. These results confirm that additional reference samples do not benefit MAGPIE; rather, they have a detrimental effect.

## C.7 BASELINE PROMPTS.

We present the system prompts used by MAGPIE-Human and MAGPIE-Few Shot for the two main target domains in the Figure 13, Figure 14, Figure 15 and Figure 16.

## C.8 POST-PROCESSING.

For the input sequences synthesized by DOMINO and other methods, we apply a unified filtering mechanism to remove low-quality samples. The filtering prompt is shown in Figure 17 and Figure 18.

```
### Characteristics of the current coding task:
1. **Problem Description**: a few sentences about the problem.

2. **Input Constraints**: Each test case has specific constraints
↪  on the size of the input, such as the number of elements in an
↪  array, the length of a string, or the range of values in an
↪  array.

3. **Output Format**: The output should match the number of test
↪  cases, with each result corresponding to the input test case.

4. **Constraints on Operations**: There may be constraints on the
↪  number of operations that can be performed, such as performing
↪  an operation at most once or within a specific range.
```

Figure 14: MAGPIE-Few Shot System Prompt for *Live Code Generation* domain.

```
### Characteristics of the current coding task:
It begins with a Python function that performs the required
↪  operations, followed by an example input provided after the
↪  function. The input example typically starts with an assert
↪  statement.
```

Figure 15: MAGPIE-Human System Prompt for *Live Code Execution* domain.

We retain only synthetic input sequences of *"Excellent"* quality. *It is important to note that the same LLM backbone used for synthesizing the domain-specific sample inputs is also used for both filtering and generating the corresponding output sequences.*

### C.9 DOMAIN ADAPTATION CONFIGURATIONS.

We fine-tuned the OPENCODER-8B Base, QWEN2.5-CODER-7B Base, and QWEN2.5-CODER-14B Base models on the synthesized domain-specific samples using LLAMA-FACTORY (Zheng et al., 2024), on a server equipped with $8\times$ NVIDIA A100 80GB GPUs. During fine-tuning, we applied LoRA (Hu et al., 2022) with `lora_rank=8` and `lora_target=all`. All experiments used the AdamW (Loshchilov & Hutter, 2019) optimizer with an initial learning rate of 5e-5. Detailed training parameters are shown in Table 9.

### C.10 A CASE STUDY OF DOMAIN-SPECIFIC SYNTHETIC SAMPLES.

From Figure 19 and Figure 20, it can be seen that the domain-specific samples synthesized by DOMINO are very similar to the domain reference samples. Although it is difficult to precisely describe this domain similarity using natural language, they exhibit consistency in certain features, such as structural characteristics.

```
### Characteristics of the current coding task:
It begins with a task description, followed by a relevant example
↪  (including both input and output) with necessary explanatory
↪  information, and ends with a set of constraints.
```

Figure 16: MAGPIE-Human System Prompt for *Live Code Generation* domain.

```
# Instruction

You need to assess the quality of the given coding question based
↪  on its clarity, specificity, completeness, and challenge.

The rating scale is as follows:

- very poor: The question is unclear, vague, or disorganized and
↪  the examples are wrong. It lacks essential information and
↪  context, contains many unreadable characters for humans or has
↪  large scale repetitive parts.
- poor: The question is somewhat unclear or lacks important
↪  details, requiring significant clarification, or contains
↪  irrelevant content and repetition.
- average: The question has moderate clarity and specificity. The
↪  question is basically complete but may require some additional
↪  information for full understanding, and the difficulty is
↪  relatively easy.
- good: The question is clear and specific, and it is generally
↪  well-articulated. The question is complete, providing
↪  sufficient context to understand its intent, with little to no
↪  irrelevant content or noise, and has a moderate level of
↪  difficulty.
- excellent: The question is very clear, specific, and
↪  well-articulated. It contains all the necessary information
↪  and context for providing a comprehensive response and also
↪  has a certain level of challenge, requiring multi-step
↪  reasoning, complex algorithms or data structures.

## Coding Question
```
{input}
```

## Output Format
Given the coding question, you first need to give an assessment,
↪  highlighting the strengths and/or weaknesses of the coding
↪  question.
Then, you need to output a rating from very poor to excellent by
↪  filling in the placeholders in [...]:
```
{{
    "explanation": "[...]",
    "input_quality": "[very poor/poor/average/good/excellent]"
}}
```
```

Figure 17: Unified filer prompt for *Live Code Generation* domain.

```
# Instruction

You need to assess the quality of the given python function based
↪  on its clarity, specificity, completeness, and challenge.

The rating scale is as follows:

- very poor: The function is unclear, vague, or disorganized. It
↪  contains many unreadable characters for humans or has large
↪  scale repetitive parts.
- poor: The function is somewhat unclear or lacks important
↪  details, contains irrelevant content and repetition.
- average: The function has moderate clarity and specificity. The
↪  difficulty is relatively easy.
- good: The function is clear and specific, and it is generally
↪  well-articulated.
- excellent: The function is very clear, specific and
↪  well-articulated, and poses a certain level of difficulty.

## Function
```
{input}
```

## Output Format
Given the function, you first need to give an assessment,
↪  highlighting the strengths and/or weaknesses of the function.
Then, you need to output a rating from very poor to excellent by
↪  filling in the placeholders in [...]:
```
{{
    "explanation": "[...]",
    "input_quality": "[very poor/poor/average/good/excellent]"
}}
```
```

Figure 18: Unified filer prompt for *Live Code Execution* domain.

Table 9: The hyper-parameters for supervised fine-tuning.

| Hyper-parameter | Value |
|---|---|
| Learning Rate | $5 \times 10^{-5}$ |
| Number of Epochs | 2 |
| Number of Devices | 8 |
| Per-device Batch Size | 3 |
| Gradient Accumulation Steps | 1 |
| Effective Batch Size | 24 |
| Optimizer | Adamw with $\beta s = (0.9, 0.999)$ and $\epsilon = 10^{-8}$ |
| Learning Rate Scheduler | linear |
| Warmup Steps | 0 |
| Max Sequence Length | 4096 |

## D  LIMITATIONS.

Our proposed method, DOMINO, encodes domain-specific patterns as minimal sufficient representations based on reference samples. To be effective, this process relies on the assumption that *the reference samples are representative of the target domain*. If the reference data is not representative of the domain, the resulting domain representation $D^*$ will fail to capture the true characteristics of the domain, ultimately limiting the quality and generalizability of the synthesized data.

You are given an integer array arr and an integer difference. Return the length of the longest subsequence in arr which is an arithmetic sequence such that the difference between adjacent elements in the subsequence is difference. A subsequence is a sequence that can be derived from another sequence by deleting some or no elements without changing the order of the remaining elements.

Example 1:
Input: arr = [1,2,3,4], difference = 1
Output: 4
Explanation: The longest arithmetic subsequence is [1,2,3,4].

Example 2:
Input: arr = [1,3,5,7], difference = 2
Output: 2
Explanation: The longest arithmetic subsequence is [1,3,5,7].

Constraints:
1 <= arr.length <= 10^5
-10^4 <= arr[i], difference <= 10^4

---

You are given an array A of length N. For each i (1 \leq i \leq N), compute the minimum value of the expression A[i] - A[j] + A[k] for all possible combinations of i, j, and k such that i \leq j \leq k.
Input
The input is given from Standard Input in the following format:
N
A_1 A_2 ... A_N
Output
Print the minimum value of the expression A[i] - A[j] + A[k] for all possible combinations of i, j, and k such that i \leq j \leq k.

Constraints
 * 3 \leq N \leq 10^5
 * 1 \leq A_i \leq 10^9
 * All values in the input are integers.

Example
Input
5
2 1 3 1 4
Output
0
For example, choosing i = 2, j = 2, and k = 3 yields the minimum value of the expression.

Input
4
4 1 5 2
Output
-4
For example, choosing i = 1, j = 2, and k = 3 yields the minimum value of the expression

Figure 19: Samples synthesized by DOMINO in the *Live Code Generation* domain.

```python
def maximumPopulation(years:
List[int]) -> int:
    n = len(years)
    cnt = [0] * 2003
    for y in years:
        cnt[y+1] += 1
        cnt[y+2] -= 1
    ans = -1
    m = 0
    for i in range(2003):
        cnt[i] += cnt[i-1]
        if cnt[i] > m:
            ans = i
            m = cnt[i]
    return ans

assert maximumPopulation(years =
[1950, 1960, 1960, 1970, 1970, 1970])
== ?
```

```python
def minimumTimeToType(word: str) -> int:
    ans = len(word)
    prev = "a"
    for w in word:
        diff = abs(ord(w) - ord(prev))
        ans += min(diff, 26 - diff)
        prev = w
    return ans

assert minimumTimeToType(word = 'a')== ?
```

Figure 20: Samples synthesized by DOMINO in the *Live Code Execution* domain.

## E    FUTURE WORK

In the methodology, while the sample-specific representations $S^{(i)}$ are discarded for synthesis, they could potentially be used for tasks like retrieving the most similar reference sample to a given query, which we leave for future work.

Further, we plan to investigate DOMINO's robustness to mixed-domain reference sets. We hypothesize that the contrastive disentanglement mechanism might naturally encourage the domain representation $D^*$ to focus on the most prevalent shared patterns, implicitly filtering out outlier noise, but this requires further study.

## F    THE USE OF LARGE LANGUAGE MODELS.

In the preparation of this manuscript, LLMs are utilized as a general-purpose writing assistant. Its role was strictly limited to improving the grammar, clarity, and readability of the text. The LLMs are not used for research ideation, conducting experiments, or the generation of any core scientific content. The authors take full responsibility for all content presented in this paper, including any text revised with the assistance of the LLM.

