# OpenReview forum: "Domain-Specific Data Synthesis for LLMs via Minimal Sufficient Representation Learning"
_ICLR.cc/2026/Conference — ICLR 2026 Conference Withdrawn Submission_

### Official Review · Reviewer_3H1n · 2025-10-29

**Soundness:** 2
**Presentation:** 3
**Contribution:** 3
**Rating:** 6
**Confidence:** 4

**Summary:**

Domino is a synthetic data generation framework that learns a minimal sufficient representation of the generation task from a reference sample. Soft tokens representing both the domain and individual samples are learned, with a contrastive loss applied to the individual sample soft tokens to encourage the domain soft tokens to be minimal. For both coding and instruction-following domains, Domino generates training data that produces better models than baseline methods. Analysis on the coding domain also demonstrates that data generated via Domino better aligns with the full distribution of reference samples than baseline methods.

**Strengths:**

- Clearly presented theory supports the formulation of the framework.
- Strong results demonstrate training base models with Domino generated data outperforms training with baseline methods, and beats the model provider’s instruction-tuned version.
- Extensive ablations and analyses to illustrate the workings and benefits of the framework.

**Weaknesses:**

- There is a bit of a disconnect between the applications pitched at the start of the Experiments section and the actual evaluation domain. The pitch is that the generation domain would be a domain whose specifications would be hard to describe in words, such as “a typical LeetCode problem from late 2024” (line 268). Instead, the actual evaluation domain was just coding in general, with the only temporal aspect being that earlier problems are used as reference examples and later examples are used as evaluation examples. If there really were temporal idiosyncrasies that could not be captured via prompt engineering, this evaluation would not capture it as the reference set would not capture the specifics of the evaluation set.
  - A potential way to demonstrate the ability of Domino to capture such specifications would be to demonstrate Domino-generated data for a particular timeframe can be used to train models that are better at problems from that timeframe than other timeframes, whereas baseline methods achieve more uniform performance across all timeframes.
- There are also synthetic data generation methods that more directly leverage the reference samples that were not baselined. While MAGPIE is a good baseline, it does not inherently utilize information from the reference samples (though the few-shot MAGPIE setting does try to). Other methods like OSS-Instruct (Wei et al., 2024) and BARE (Zhu et al., 2025) utilize reference examples directly in the prompt to ground synthetic data generation and would be worth comparing against.
- It is unclear whether the generation methods are compared holding training data size constant or compute constant. The authors mention that a filtering step was applied to the generated dataset for all methods, but I could not find discussion on how large the filtered dataset was for any method. If the filtering process was biased towards retaining synthetic data generated by Domino, this would mean more training on the Domino-generated dataset and thus better performance.
  - Similarly, I feel including the reference SFT row in the results tables is unnecessary as much less training would have been performed on the much smaller reference set, unless you were to increase the number of training epochs to bring the number of training steps to parity with the other methods.
- Confidence intervals are not provided for the results, which makes it difficult to contextualize comparisons against baseline methods. Given the size of the evaluation sets (≤ 200) and the small scale of the performance differences, it would be helpful to have an idea of the uncertainty in the metrics.

**Questions:**

- Do you have insights into why not including sample soft tokens at generation time works well vs, e.g., including dummy sample soft tokens?
- For the samples described in Table 2, was the sample presented representative of all samples from D? Line 407 mentions how the samples in the table from $\mathbf{D}^*$ were selected specifically to be far from the “the dense cluster representing $\mathbf{D}$,” but I don't think it is shown how dense that cluster is. Moreover, is the reference sample in the table the closest one to the sample from $\mathbf{D}$? I would just like more clarification on whether Table 2 should be interpreted as a case study of a single sample or indicative of a general issue with data diversity when using Domino with only the $\mathcal{L}_1$ loss.
- The ablations on $\lambda$ focus only on the code generation task. Do you have insights into whether the ablation recommendations generalize to other tasks, or would the choice of lambda be task-specific?

---

### Official Review · Reviewer_NMuv · 2025-11-01

**Soundness:** 2
**Presentation:** 2
**Contribution:** 2
**Rating:** 2
**Confidence:** 3

**Summary:**

This work proposes DOMINO, which generates domain-specific synthetic data using example samples, without requiring explicitly defined descriptions. It learns a minimal sufficient domain representation via prompt tuning and contrastive disentanglement to capture core domain patterns and avoid overfitting to sample-specific details. Experiments show that DOMINO improves performance on coding and instruction-following tasks compared to the baselines.

**Strengths:**

The motivation to generate synthetic data for domains that are difficult to explicitly describe is interesting. The proposed approach also shows good performance compared to baselines. It provides extensive visualizations and analysis results that illustrate the behavior of the proposed method.

**Weaknesses:**

The method could benefit from comparisons with a broader range of state-of-the-art synthetic data generation baselines, and the evaluation metrics could be expanded to more comprehensively assess both data quality and downstream performance.

It is also not very clear how closely the synthesized data resemble the reference domain or how they compare to outputs generated by existing SOTA approaches. Providing direct qualitative comparisons would make the evaluation more convincing (e.g., showing samples from Figs. 19 and 20 alongside reference data and samples from other baselines).

The method appears highly dependent on the quality and representativeness of the reference samples. How reference examples are selected and how robust the approach is to noisy or imperfect input?

There are also minor presentation issues that affect readability, including inconsistent table font sizes, overlapping text in Fig. 5, and the yellow line in Fig. 3 being difficult to discern. Addressing these points would further enhance clarity.

**Questions:**

It would be helpful to include a performance curve showing how model performance changes as the amount of generated synthetic data increases. Presenting this trend alongside other baselines would allow a clearer comparison of data efficiency.

Similarly, I would like to see how performance varies with respect to the number of reference samples. Understanding sensitivity to reference set size would provide more insight into practical applicability.

What is the exact definition of the Pass@1 metric used in the experiments? A brief clarification in the main text would help ensure consistent interpretation.

---

### Official Review · Reviewer_tYar · 2025-11-01

**Soundness:** 3
**Presentation:** 3
**Contribution:** 3
**Rating:** 6
**Confidence:** 4

**Summary:**

The paper presents DOMINO, a framework for learning minimal sufficient domain representations from reference samples and using them to generate domain-specific synthetic data. Current synthetic data generation methods depend heavily on human-defined domain descriptions expressed in natural language. DOMINO, on the other hand, learns a set of domain soft tokens directly from reference examples through an inductive approach and uses these tokens as input to guide LLMs in generating synthetic data.
This is achieved through prompt tuning using a combined loss function consisting of domain-level likelihood (L₁) and contrastive loss (L₂). The contrastive component leverages sample-specific representations to enforce disentanglement between shared domain patterns and individual sample details, preventing overfitting while preserving core domain characteristics. The authors provide theoretical proofs demonstrating that their approach learns representations with both sufficiency and minimal sufficiency properties, and prove that the method expands the support of the synthetic data distribution compared to standard prompt tuning.
The paper uses LiveCodeBench as the primary testbed, demonstrating performance improvements of up to 4.63% in Pass@1 accuracy when fine-tuning on DOMINO-synthesized data compared to strong instruction-tuned baselines.

**Strengths:**

[1] The DOMINO framework addresses a critical limitation in current data synthesis methods by eliminating the need for manual prompt engineering and explicit domain descriptions. Instead, it learns minimal sufficient domain representations directly from reference examples through an inductive approach, making it applicable to domains that are difficult to articulate in natural language. [2] The paper provides rigorous mathematical foundations for domain-specific data synthesis through information-theoretic concepts of sufficiency and minimal sufficiency. The theoretical guarantees demonstrate that DOMINO learns representations that capture essential domain characteristics while discarding irrelevant sample-specific noise, with formal proofs showing expanded support of the synthetic data distribution. [3] The authors demonstrate DOMINO's generalization capability by extending beyond coding tasks to diverse instruction-following domains. Results show consistent improvements across these varied task types, validating the framework's robustness and broad applicability beyond its primary coding benchmark evaluation.

**Weaknesses:**

1. Synthetic Data Quality Issues: The examples provided reveal significant correctness problems that undermine the method's reliability. In Figure 1, the synthetic sample incorrectly identifies subarrays [2,3] and [3] as valid when they sum to odd numbers, violating the problem's even-sum requirement. Similarly, Figure 9 shows fundamental logical errors where the explanation correctly identifies the need to count distinct subsequences but then incorrectly counts "aba" and "baba" multiple times instead of once. These are not minor formatting issues but quality failures that question whether DOMINO preserves problem constraints during synthesis.

2. The paper's central claim of increased diversity lacks convincing quantitative support. Figure 3 shows DOMINO samples more tightly clustered around reference points for Live Code Generation, suggesting reduced rather than increased diversity. This aligns with the weaker performance gains in Table 1 for this domain compared to Live Code Execution where Figure 4 shows more dispersed samples. The authors need more rigorous diversity metrics beyond t-SNE visualizations and should address this apparent contradiction between claimed and observed diversity patterns.

3. The uniform prior assumption p(D)∝1 is fundamentally violated when using code-specialized LLMs on coding benchmarks, as these models already possess substantial domain knowledge. The temporal cutoff in LiveCodeBench doesn't guarantee that coding paradigms and problem patterns haven't leaked from test to training data, making evaluation potentially inflated. The authors should evaluate on truly held-out domains or provide stronger guarantees about train-test separation to validate performance gains using DOMINO.

**Questions:**

1. How can data quality be verified/addressed within the DONIMO framework ? As mentioned in the point [1] in weakness above the generated samples seem to have logical inconsistencies within them that will have negative impact when used for finetuning candidate models.
2. For showing diversity comparison authors should include quantitative metrics other than just t-SNE plots which is more visual. Also, diversity is difficult to capture with just one metric and hence should be computed across a mirad of metrics such as compression ratio, Remote clique, chamfer distance, etc. Please refer - METASYNTH: Meta–Prompting–Driven Agentic Scaffolds for Diverse Synthetic Data Generation (https://arxiv.org/pdf/2504.12563) for the full list of diversity metrics which is a prompt driven agentic approach to synthetic data generation.  Additionally, how do these diversity measures correlate with downstream performance improvements to establish whether increased diversity actually translates to better model training?
3. In my opinion a low resource domain is a must to demonstrate the effectiveness of the DOMINO framework as it will gaurantee no prior knowledge of the domain. My concern is emanating from the observation made in point 3 in weakness. Could the authors demonstrate DOMINO's effectiveness on truly low-resource domains where the backbone LLM has minimal prior knowledge? This would provide stronger evidence for the framework's ability to learn domain representations from scratch rather than refining existing knowledge.
4. In Appendix D: Limitations you correctly point out that " To be effective, this process relies on the assumption that the reference samples are representative of the target domain. " Doesn't this shift the burden from crafting good natural language descriptions to curating high-quality exemplars that span the domain distribution? This again becomes non-trivial for low resource/obscure domains and one again has to take the path of collecting as many samples as possible to feed as reference to cast a wider net to capture the domain distribution, baring which you will only be able to generate limited samples similar to the reference samples (this seems to be the case in Figure 3)

---

### Author Response · Authors · 2025-11-22
**Statement by Authors**

We sincerely thank all the reviewers for taking the time to review our paper. Your encouraging comments on the novelty and intuitiveness of our core idea, as well as your recognition of the comprehensive and rigorous evaluation, are deeply appreciated. Regrettably, owing to insufficient computational resources, we are unable to provide the additional experimental results requested. Therefore, we have decided to withdraw the manuscript.

---

### Note · Authors · 2026-01-05

**Comment:**

We sincerely thank all the reviewers for taking the time to review our paper. Your encouraging comments on the novelty and intuitiveness of our core idea, as well as your recognition of the comprehensive and rigorous evaluation, are deeply appreciated. Regrettably, owing to insufficient computational resources, we are unable to provide the additional experimental results requested. Therefore, we have decided to withdraw the manuscript.

**Withdrawal Confirmation:**

I have read and agree with the venue's withdrawal policy on behalf of myself and my co-authors.